# The Effects of Cross-Linking Agents on the Mechanical Properties of Poly (Methyl Methacrylate) Resin

**DOI:** 10.3390/polym15102387

**Published:** 2023-05-19

**Authors:** Gulsum Ceylan, Serkan Emik, Tuncer Yalcinyuva, Emin Sunbuloğlu, Ergun Bozdag, Fatma Unalan

**Affiliations:** 1Department of Prosthodontics, School of Dentistry, Istanbul Medipol University, Istanbul 34083, Turkey; 2Department of Chemical Engineering, Faculty of Engineering, Istanbul University-Cerrahpasa, Istanbul 34320, Turkey; 3Department of Mechanical Engineering, Laboratory of Biomechanics & Mechanics of Materials, Faculty of Engineering, Istanbul Technical University, Istanbul 34437, Turkey; 4Department of Prosthodontics, Faculty of Dentistry, Istanbul Kent University, Istanbul 34433, Turkey

**Keywords:** denture base polymers, cross-linking agent, flexural strength, impact strength, surface hardness

## Abstract

Cross-linking agents are incorporated into denture base materials to improve their mechanical properties. This study investigated the effects of various cross-linking agents, with different cross-linking chain lengths and flexibilities, on the flexural strength, impact strength, and surface hardness of polymethyl methacrylate (PMMA). The cross-linking agents used were ethylene glycol dimethacrylate (EGDMA), tetraethylene glycol dimethacrylate (TEGDMA), tetraethylene glycol diacrylate (TEGDA), and polyethylene glycol dimethacrylate (PEGDMA). These agents were added to the methyl methacrylate (MMA) monomer component in concentrations of 5%, 10%, 15%, and 20% by volume and 10% by molecular weight. A total of 630 specimens, comprising 21 groups, were fabricated. Flexural strength and elastic modulus were assessed using a 3-point bending test, impact strength was measured via the Charpy type test, and surface Vickers hardness was determined. Statistical analyses were performed using the Kolmogorov–Smirnov Test, Kruskal–Wallis Test, Mann–Whitney U Test, and ANOVA with post hoc Tamhane test (*p* ≤ 0.05). No significant increase in flexural strength, elastic modulus, or impact strength was observed in the cross-linking groups compared to conventional PMMA. However, surface hardness values notably decreased with the addition of 5% to 20% PEGDMA. The incorporation of cross-linking agents in concentrations ranging from 5% to 15% led to an improvement in the mechanical properties of PMMA.

## 1. Introduction

Polymethyl methacrylate (PMMA) has been extensively used in dentistry for many years to fabricate removable denture bases [1,2]. It is a cost-effective, aesthetically appealing, and easy-to-use material. However, it has low impact and fatigue resistance, among the most important disadvantages. Prosthetic fractures are frequently seen due to the low impact and fatigue resistance of PMMA [3].

Prosthetic fractures typically occur following bending fatigue. Flexural strength (FS) represents the maximum stress a material can withstand before yielding or breaking in a flexural test. Impact strength (IS), which may arise from accidentally dropping the prosthesis [4], is defined as a material’s ability to resist sudden applied force or stress. Repeated intraoral forces contribute to the development of microcracks in the denture base material, eventually causing denture fractures due to bending fatigue or impact forces [5].

Three primary methods prevent prosthetic fractures: creating a new alternative to PMMA, reinforcing PMMA with various fibers (such as carbon, aramid, glass, and polyethylene) or metal, and modifying PMMA’s chemical structure [6]. Chemical modification of PMMA involves adding a diverse range of cross-linking agents to the resin or altering the resin’s chemical structure through copolymerization with elastomers [7]. Strengthening glassy polymers with elastomers is a well-researched and accepted approach. In addition, elastomeric or high-impact resins can absorb more energy than conventional resins [8,9].

In recent years, efforts have been made to improve the properties of denture base resins through monomeric modifications. For instance, the MMA monomer has been modified by adding fluoromonomers, phosphate monomers, methacrylic acid monomers, itaconia monomers, nitro monomers, or other non-specific monomers or solutions [10]. Although these modifications were intended to enhance the mechanical properties of denture base polymers, most studies have yet to provide definitive reports on the copolymerization of these modifications with PMMA [10,11,12,13].

A cross-linking agent is a monomer featuring two or more groups per molecule capable of polymerization. Each active group can be integrated into a growing polymer chain during polymerization, forming a ring or crosslink between two chains [14]. The primary objective of incorporating a cross-linking agent into PMMA content is to enhance the material’s resistance to cracking [15]. The cross-linking agent imbues two crucial properties to the polymerized material: reducing the denture base’s dissolution in organic solvents and increasing its resistance to stress formation [4]. Cross-linking agents are added to the liquid portion of the denture base material to improve impact strength, flexural strength, and surface hardness. Numerous cross-linking agents can be incorporated into the PMMA content [14], with their most significant advantage being the resultant polymer’s resistance to small surface cracks [9].

The effect of cross-linking agents on denture base material properties has been examined extensively. Some studies report that adding high percentages of cross-linking agents may reduce the material’s mechanical properties, such as flexural strength, impact strength, and surface hardness. Thus, the proportion of cross-linking agents added to the liquid fraction is essential [16]. The chain length of the cross-linking agent in the monomer content also plays a vital role, as it influences the formation of different chemical structures. Cross-linking agents with varying chain structures and lengths can be added individually or combined with other agents to the monomer content for this purpose [17].

In modern dentistry, CAD/CAM technology is increasingly indispensable for removable prostheses. CAD/CAM denture bases offer several advantages, including reduced clinical chair time and fewer patient visits. However, long-term clinical outcome studies are needed to confirm their effectiveness. Therefore, the quest for new materials to enhance denture base properties continues [18,19].

This study aims to evaluate the effects of cross-linking agents on the mechanical properties of PMMA. The objectives of this study are to determine the optimal concentration of cross-linking agents for improving the mechanical properties of PMMA and compare the mechanical properties of PMMA with and without cross-linking agents. The null hypothesis states that incorporating cross-linking agents with varying chain lengths and in different percentages does not significantly affect the elastic modulus, flexural strength, impact strength, and surface hardness of PMMA denture base materials.

## 2. Materials and Methods

### 2.1. Specimen Preparation

Metal samples prepared from stainless steel in rectangular prisms of 3.3 × 10 × 64 mm for three-point bending and surface hardness test specimens and 4 × 6 × 50 mm for impact strength test specimens were used to form the mold cavities. The traditional molding methods and brass muffles with spring brits were used to prepare acrylic resin samples. First, type II hard gypsum (Moldano, Heraeus Kulzer, Ltd., Hanau, Germany) was mixed according to the manufacturer’s instructions. Next, gypsum was filled into the lower part of the muffles, and then rectangular prism-shaped specimens were Hanauplaced on the gypsum. After the gypsum hardened, the gypsum surface was sanded with 600 Grid sandpaper to ensure smoothness of the surface. The gypsum surface was then isolated using an alginate-based sealant (Isolant Separating Solution, Dentsply Corp., Dentsply, DeTrey, UK). After all these procedures, the upper part of the muffle was placed and filled with hard gypsum, the upper cover of the muffle was closed, taken to a hydraulic press (Kavo Elektrotechnisches Werk GmBH, Germany), and kept under 2 atm pressure for 2 h. After the curing reaction of the gypsum was completed, the sample molds were removed from the muffle. Then, both gypsum surfaces of the muffle were isolated using a thin layer of alginate-based isolating material.

The acrylic resin, monomer, and cross-linking agents used in this study are presented in Table 1.

Four types of cross-linking agents were evaluated: EGDMA, TEGDA, TEGDMA, and PEGDMA. The cross-linking agents were added to the MMA monomer component in the following concentrations: 5%, 10%, 15%, 20% of the volume, and 10% of the molecular weight. Acrylic resin powder and liquid were mixed at a ratio of 23.4 mg powder per 10 mL liquid according to the manufacturer’s instructions. Monomers for groups using cross-linking agents by volume were prepared as follows: for 5%, 0.5 mL cross-linking agent-9.5 mL MMA, for 10%, 1 mL cross-linking agent-9 mL MMA, for 15%, 1.5 mL cross-linking agent-8.5 mL MMA, and for 20%, 2 mL of cross-linking agent-8 mL of MMA.

Due to the varying molecular weights of the cross-linking agents, the chain lengths also differ. Given that long-chain cross-linking agents will produce fewer bonds by volume compared to short and straight-chain cross-linking agents, an experimental group containing 10 moles of the cross-linking agent was formed to evaluate the effect of only one molecular weight. The schematic representation of the structures synthesized in the study and how the crosslinker is attached to the main polymer structure is given in Figure 1. In preparing the 10% monomer by molecular weight, 1 mole of the cross-linking agent was combined with 9 moles of MMA. Based on these ratios, monomers were prepared as 1 mL EGDMA-9 mL MMA, 1.76 mL TEGDMA-9 mL MMA, 1.9 mL TEGDA-9 mL MMA, and 2.6 mL PEGDMA-9 mL MMA.

The amount of monomer was adjusted with a scale and poured into the glass container. The polymer weighed in the specified amount on a precision balance and was slowly added into the monomer. After the mixture was stirred for one minute, it was covered with glass to prevent the loss of the monomer in its content, and the doughing period of 6 min was expected to be completed. At the end of this period, the acrylic dough was placed in previously prepared muffle molds. After it was covered with the other part of the muffle, it was placed in a hydraulic press, compressed under 2 atm pressure and kept in the press for 4 h. The screws of the muffles taken from the hydraulic press were tightened well to prevent the muffle from rising.

In the polymerization processes, the heat polymerization method in water was used. First, the samples were placed in the polymerization device containing water at room temperature (Kavo EWL Typ 5506, Kavo Elektronisches Werk GmBH, Allgäu, Germany) following the manufacturer’s instructions and kept for 30 min until the temperature of the water reaches 70 °C. It is maintained for 30 min at 70 °C, and then the device adjusted to 100 °C. The water was boiled for 30 min after boiling, and the acrylic was allowed to polymerize. After the polymerization process, the device was turned off. Then, the muffles were left in water for 24 h until the water in the device cooled utterly, and the samples whose polymerization was completed were carefully removed from the muffle.

The surfaces of the test specimens were sanded with 320, 400, 600, and 1200-grit sandpaper to ensure complete smoothness. The polishing process continued until the sample surfaces reached the ideal level of brightness.

For this study, a total of 630 specimens were prepared, comprising 420 specimens for three-point bending and surface hardness tests and 210 specimens for impact strength tests. All the samples prepared for the experiments were stored in a 3 °C incubator (Memmert Model 600, Schwabach, Germany) in distilled water for 48 h. Before the experiments, samples were removed from distilled water and allowed to acclimate at room temperature for 1 h according to ISO 1567 [20].

### 2.2. Flexural Strength Test

Ten rectangular specimens from each group measuring 64 × 10 × 3.3 mm were prepared as specified by the International Standards Organization (ISO) specification [20]. The flexural strength test was performed using a 3-point bending test using a universal testing machine (Mini Bionix II, MTS Systems Corporation, Eden Praire, MN, USA) calibrated with a 500 kg load cell and a crosshead speed of 5 mm/min. The flexural testing device consisted of a central loading plunger and two polished cylindrical supports, 3.2 mm in diameter and 10.5 mm long. The distance between the centers of the supports was 50 mm. The compressive force was applied perpendicular to the center of the specimens until a deviation in the load-deflection curve and specimen fracture occurred. The flexural strength and modulus were recorded in megapascal (MPa), and the maximum force was recorded in Newton (N). The ultimate flexural strength and elastic modulus were calculated using the following formulas: FS = 3F/2bh2, where: FS = flexural strength; F = load at fracture (N); L = distance between the supports of the sample (50 mm); b = sample width (10 mm); and h = sample thickness (3.3 mm). E = FL3/4bdh3, where: E = elastic modulus; F = load in some point of the linear region of the stress-strain curve (N); L = distance between the supports of the sample (50 mm); b = sample width (10 mm); h = sample thickness (3.3 mm); and d = slack compensated deflection at load (F).

### 2.3. Impact Strength Test

Ten rectangular specimens from each group measuring 50 × 6 × 4 mm were prepared according to the British Standards for the Testing of Denture Base Resins (BS 2487: 1989) and British Standard Specification for Orthodontic Resins (BS 6747: 1987) [21,22]. A type V notch was cut in the middle of each specimen using a notch cutter (Model CNB35-001A1, Blacks Equipment Ltd., Doncaster, England) and a V-shaped milling tool. The depth of the notch was 0.8 mm on the 4 mm surface of the specimen and was controlled using the Optic Profile Projector Device (Jena model 1087, Carl Zeiss Jena, Leipzig, Germany) with 50× magnification.

The impact strength was determined with an Impact Testing Machine (Akagun Engineering, Istanbul, Turkey) using the Charpy method and a 4 J pendulum. The specimens were horizontally positioned with a distance of 40 mm between the fixed supports. Samples were placed on horizontal supports with the midpoint in the pendulum’s path. The pendulum was released from the rest position, and the reduction in the pendulum’s swing immediately after breaking the specimen was indicated by the pointer’s position on the attached dial scale. The direct reading of the scale multiplied by the pendulum’s weight provided a value which was then converted to kJ using the manufacturer’s conversion chart. The amount of energy read for each sample was read from the instrument’s display.

The calculated air friction values were then subtracted from these energy amounts. Finally, the impact resistance for each sample was calculated by dividing the surface area of the samples in kJ/m^2^ using the following formula:

IS = EC/hbA, where: IS = impact strength; EC = amount of energy absorbed; bA = residual thickness from the notch (4-08 = 3.2 mm); and h = height of the sample (6 mm).

### 2.4. Surface Hardness Test

For the surface hardness test, samples prepared in dimensions of 64 × 10 × 3.3 mm were shortened to 30 ± 2 mm in diameter and 20 ± 2 mm in height to fit into cylindrical molds, as the sample surface was easy to polish in the polishing machine and suitable for the measuring device. Autopolymerizing acrylic resin in a flowing consistency was poured onto the test sample and placed on the bottom of the cylindrical molds. The surface of the test specimen was first re-sanded with 320, 400, 600, and 1200-grit sandpaper to ensure that the surface was completely smooth. Then, the polishing process with polishing paste was carried out with the polishing machine until the sample surface reached the ideal brightness. For surface hardness measurement, care was taken to ensure that the bottom and top surfaces of the test specimens were parallel to each other so that the tip could penetrate perpendicular to the test specimen’s surface, ensuring accurate measurements.

A Vickers Microhardness Tester (Shimadzu HMV-2L; Shimadzu Corp., Kyoto, Japan) was used to determine the surface hardness. First, the samples were placed in the instrument, and the microscope image of the device was activated, providing a clear view of the surface where the hardness-measuring tip penetrated. Then, the hardness meter tip was started, and the force applied to the sample surface was set to 25 gf for 30 s. Next, the device retracted the tip, which had penetrated the sample for 30 s. Finally, the diagonal shape on the sample surface was marked with a microscope, and the instrument measured the hardness. Measurements were made at three points on each test sample, and the surface hardness data were recorded for each sample by calculating the arithmetic mean of the three measurements. The Vickers hardness number for surface hardness was calculated automatically by the tester using the following formula: VHN = 1.8544 P/d^2^, where: VHN = Vickers Hardness Number; P = force applied to the surface; and d = the diagonal length of the tip applied to the surface.

### 2.5. Statistical Analysis

SPSS 16.0 (Statistical Package for the Social Sciences) software was used for statistical analysis. To evaluate the study data, the Kolmogorov–Smirnov test was used to check if the data met the assumptions of parametric tests. If the assumptions were not met, the Kruskal–Wallis test was used. In case of differences in Kruskal–Wallis Test results, the Mann–Whitney U test was used in binary comparisons. If the ANOVA test results were not equal, the post hoc Tamhane test was used. The results were evaluated at a *p* ≤ 0.05 significance level.

## 3. Results

Definitions of measurement variables and mean values for all tests are shown in Table 2.

The elastic modulus values were significantly different between the groups (*p* ≤ 0.01): EGDMA (*p* = 0.001), TEGDMA (*p* = 0.002), TEGDA (*p* = 0.001), and PEGDMA (*p* = 0.0001). The highest mean value was found in the EGDMA 15% group (3378.06 ± 195.915), and the lowest mean value was found in the PEGDMA 10% group (2592.52 ± 299.765) (Table 3, Figure 2).

The flexural strength data were significantly different between the groups (*p* ≤ 0.05) except for the TEGDMA group (*p* = 0.067). The highest mean value was found in the TEGDA 15% group (122.26 ± 9.051 MPa), and the lowest mean value was found in the EGDMA 20% group (88.51 ± 7.252 MPa) (Table 3, Figure 3).

No statistically significant difference was found between the groups’ impact strength values (*p* ≤ 0.05). However, the highest mean value was found in the control group (6.49 ± 0.384), and the lowest mean value was found in the TEGDAm 10% group (5.80 ± 0.373) (Table 3, Figure 4).

The surface hardness values were significantly different between the TEGDMA and PEGDMA groups (*p* = 0.035 and *p* = 0.007, respectively). No statistically significant difference was found between the EGDMA and TEGDA groups (*p* = 0.161 and *p* = 0.933, respectively). The highest mean value was found in the TEGDA 20% group (17.95 ± 1.156 VHN), and the lowest mean value was found in the PEGDMA 10% group (14.43 ± 0.814) (Table 3, Figure 5).

## 4. Discussion

Modifications of the monomer content of denture base acrylic resins have often been attempted. Numerous studies have investigated using monomer modifications to evaluate the interaction of mechanical properties of denture base resins [10]. Cross-linking agents, such as EGDMA and 1,4-butylene glycol dimethacrylate (Bis-GMA), are commonly added to PMMA. These cross-linking agents provide benefits by reducing PMMA’s tendency to dissolve in organic solvents and improving its resistance to crazing [23]. In addition, cross-linkers diminish the formation of oxygen inhibition layers and residual monomers in polymerized materials, resulting in denture base materials demonstrating satisfactory chemical stability in the oral cavity [1]. The present study evaluated the effect of different cross-linking agents added to PMMA at various concentrations on mechanical properties. The presented data provide insights regarding the effectiveness of adding cross-linking agents with varying chain lengths and ratios into the PMMA monomer component. Based on the study’s statistical analysis, the null hypotheses were rejected, as significant differences between the groups in terms of elastic modulus, flexural strength, and surface hardness were observed.

According to ISO 1567 denture base materials specification, flexural strength should be ≥65 MPa and impact strength should be 2 kJ/m [20]. Therefore, the elastic modulus, flexural strength, impact strength, and surface hardness findings of all groups are in accordance with the determined mechanical property criteria of prosthetic denture base materials. Flexural strength, also known as modulus of rupture, bend strength, or transverse rupture strength, is a material property defined as the stress in a material just before it yields in a flexure test. Since a denture base may fracture in real life for various reasons, its material must have high flexural strength. Findings related to the flexural strength of denture base materials with added cross-linking agents vary. The concentration of the cross-linking agents in the monomer is usually limited to not more than 15% [17]. At this level of cross-linking, the polymer’s susceptibility to solvent crazing is significantly reduced but not eliminated [15]. Higher concentrations of cross-linking agents are avoided because of their observed effect in reducing tensile strength and impact resistance. In the present study, flexural strength and elastic modulus values decreased when the cross-linking agent was used at the rate of 20% in all groups. This result was consistent with previous reports [24,25]. Therefore, adding a maximum of 15% cross-linking agent to the monomer content is recommended, as the material’s mechanical properties, such as flexural strength and modulus of elasticity, will decrease if a high amount of cross-linking agent is added.

Dimethacrylate monomer is added to form a highly cross-linked polymer, resulting in a three-dimensional complex structure [26,27]. The EGDMA cross-linking agent is often used because of its short chain structure, which means less risk of residual non-polymerized monomer [28]. The best flexural strength value was obtained in the present study using the EGDMA15% group. This result can be attributed to the chain structure of the EGDMA group and the use of 15% of the content of the mixture, as suggested in many studies [24,29].

The addition of materials such as metal or fiber in various forms is frequently used to reinforce the acrylic denture base material. However, since these materials do not form a chemical connection with the denture base material, they may be mechanically and aesthetically disadvantageous [30]. Therefore, it is advantageous that adding a cross-linking agent does not create an aesthetic and functional problem with the material.

Fractures caused by dropping prostheses on a hard surface are frequently seen, especially in elderly patients with systemic joint and muscle disorders. Impact resistance is the amount of energy absorbed by the material before breaking. Materials with good impact resistance readily absorb energy due to their elastomeric behavior. Therefore, the flexibility of the material is an important feature. PMMA is a hard and brittle material in the oral cavity at physiological temperature. Thus, the impact resistance of PMMA is relatively low [15,31].

Many studies in the literature examine the impact strength of denture base materials. Factors impacting resistance depend on sample sizes, presence and depth of notches, loading configuration (Charpy or Izod), and pulse velocity. When studies on the impact resistance of acrylic resins are examined, the Charpy test is frequently used [28,32]. The Charpy installation configuration is also used in this study. In the case of standards for assessing the material’s impact resistance, the sample standard is notched when the Charpy loading type is used. In the Charpy impact test, notched samples are often used in the literature [6,33]. Notched test specimens are recommended to mimic the frenulum, a stress zone in the Charpy-type impact strength test [25]. However, it has also been reported that stress can accumulate in the material during the preparation of the notches in the test specimens [34]. Studies comparing the impact strength of notched and unnotched samples have reported that the same values were observed in both groups [4].

Impact strength may decrease with increasing concentrations of cross-linking agents with difunctional and trifunctional groups and chain lengths [35,36]. This is similar to the results of this study, which found that impact strength generally decreased with increasing concentrations of cross-linking agents and increased with lower concentrations.

EDGMA cross-linking agent affects the material’s wear and impact resistance. The reaction of the cross-linking agent is complex, involving the degree of polymerization, the glass transition temperature (Tg) of the PMMA, the geometric constraints of the polymer, and the presence of an unreacted cross-linking agent in the material, which serves as a residual monomer or plasticizer [37].

Long-chain cross-linking agents make the material more flexible and reduce the flexural strength and modulus of elasticity. The high concentration of long-chain PEGDMA has been reported to cause a decrease in the material’s mechanical properties. In contrast, shorter chain TEGDMA, compared to PEGDMA, results in fewer adverse effects on the mechanical properties of the substance at higher concentrations [9,38]. These results are similar to this study. PEGDMA cross-linking agent, with a more extended chain structure, has been noted in many studies to require higher polymerization temperature and pressure to polymerize the added groups. In this study, a classical molding technique was used, and it can be thought that the cross-linking reaction is complex. Polymerization temperatures below the Tg of PEGDMA groups may cause a decrease in mechanical properties. As a residual monomer or pendant chain, the unreacted cross-linking agent will act as a plasticizer. In other studies, researchers evaluated the effect of the chain length of cross-linking agent on the mechanical properties of the resulting composite structure. It has been reported that using long-chain cross-linking agents significantly increases the impact resistance of PMMA compared to short-chain EGDMA [16]. This study obtained high-impact resistance values in PEGDMA-containing groups. However, the obtained values were not statistically different from the other groups.

Surface hardness is an essential factor that affects the material’s physical properties in response to occlusal forces and mechanical prosthesis cleaning in the mouth [39]. In the literature, there is no standard for test specimens when evaluating the surface hardness of prosthetic base materials, and samples and tests of different sizes are used. In the literature, 64 × 10 × 3 mm or disc-shaped samples are frequently used [17,39].

Different polymerization methods have shown that light-activated acrylic resin has higher surface hardness than hot acrylic and auto-polymerizing acrylic resin [40]. This indicates that the mechanical and physical properties of the material may change in different polymerization techniques. In this study, different results may be obtained with varying boiling times because the short-term boiling method was applied as recommended by the manufacturer.

Regardless of the method used in the polymerization reactions of acrylic resins, the conversion of the monomers to the polymer is incomplete, and the polymerized resin retains varying amounts of free or unreacted monomer [41]. Residual monomer acts as a plasticizer in the polymer matrix, causing porosities that affect acrylic resins’ physical and mechanical properties [42,43]. In particular, to fully react to the cross-linking agent contained in the monomer content, it is necessary to provide high-temperature polymerization conditions [43]. When different polymerization methods are applied, such as long and short boiling, the structure of the polymer formed via the microwave polymerization method and the amount of residual monomer may vary [2]. Therefore, we believe it is possible to achieve lower values in the experimental groups where cross-linking agents are used compared to the control group, in the polymerization device with a thermostat, and in an environment in which the maximum temperature of 100 °C can be achieved.

Polyamide denture base material, known for its elasticity, has fewer cross-linking agents that can affect surface hardness [44]. In addition, 3D acrylic resins used to make removable dentures have relatively low double bond conversion compared to conventional acrylic resins, which can also affect mechanical properties [45]. In contrast, with CAD/CAM resin, high pressure affects the formation of longer polymer chains and can lead to a higher degree of monomer conversion [46]. Furthermore, the variety and amount of inorganic fillers in the polymerization process of CAD/CAM resins and the high temperatures applied during polymerization improve some mechanical properties, such as flexural strength and surface hardness [18,43].

The properties of PMMA materials can be enhanced through chemical modifications, such as incorporating rubber to create a PMMA–rubber network that increases impact strength [1]. This modification slows crack propagation and improves fatigue resistance, providing dentures with additional flexibility. Rubber–PMMA material properties change with rubber concentration, affecting Young’s modulus and tensile strength values [23]. Rubber-incorporated PMMA is beneficial for patients prone to dropping their prosthesis but is more expensive [1,23]. Limited research exists on PMMA chemical modification, warranting further studies with various materials, including cross-linkers, resins, and copolymers.

One of the significant limitations of this study is that there are few recent studies on this subject, which limits comparison with previous research. Furthermore, this study has additional limitations. Materials were tested under laboratory conditions, but in clinical settings, their properties may differ from those observed. In addition, the conventional polymerization technique was used, but other methods could be employed to complete the polymerization of cross-linking agents. To address these limitations, further research should involve closely simulated clinical situations and explore a wider range of acrylic resin materials and cross-linking agents to test and compare mechanical properties. SEM analysis to evaluate porosity and phase separations in experimental samples could also help determine the optimal cross-linking agent ratio.

## 5. Conclusions

The addition of cross-linking agents to conventional PMMA did not significantly increase flexural strength, elastic modulus, and impact strength values. This suggests that incorporating cross-linking agents within the concentrations and chain lengths studied did not substantially improve these mechanical properties. Interestingly, adding PEGDMA at 5% to 20% concentrations significantly decreased the surface hardness values. This finding indicates that higher concentrations of PEGDMA negatively affected the surface hardness of the PMMA denture base material. However, when considering the overall effects of cross-linking agents, it was observed that adding these agents at concentrations between 5% and 15% resulted in improved mechanical properties of conventional PMMA denture base material. Through observing these results and selecting the right cross-linker, tuning the mechanical properties in the desired direction is possible.

## Figures and Tables

**Figure 1 polymers-15-02387-f001:**
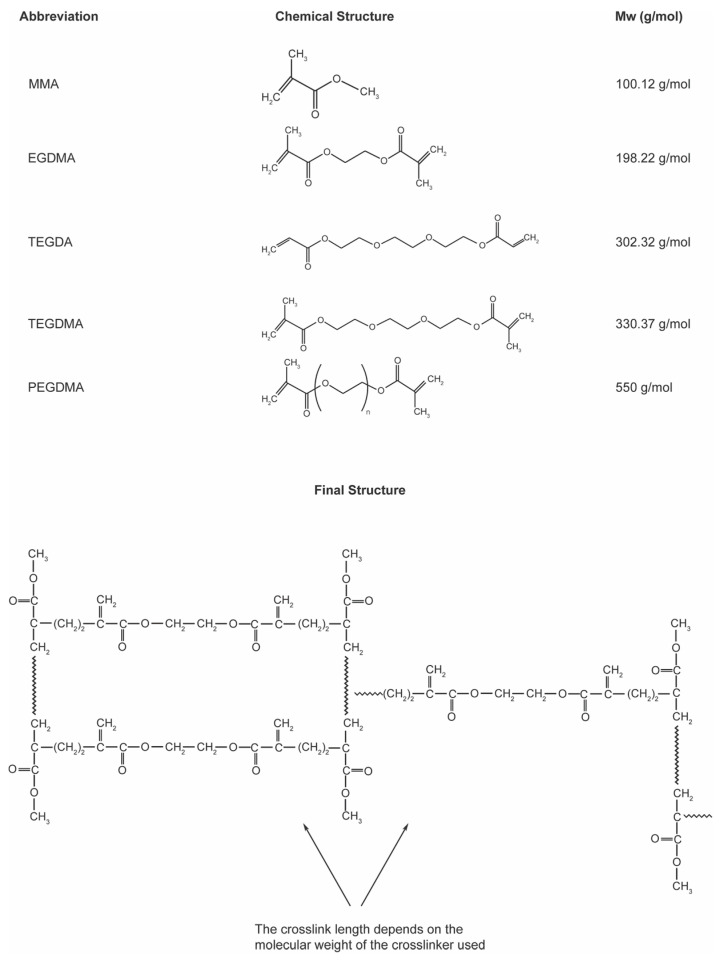
The schematic representation of structures synthesized in the study and how the crosslinker is attached to the main polymer structure.

**Figure 2 polymers-15-02387-f002:**
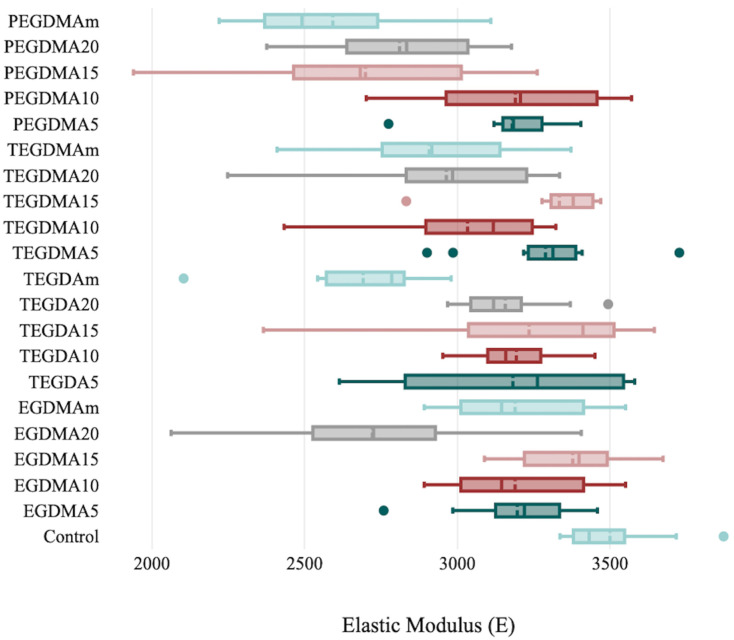
Elastic modulus evaluation parameters and standard deviation bars for the control and cross-linking groups.

**Figure 3 polymers-15-02387-f003:**
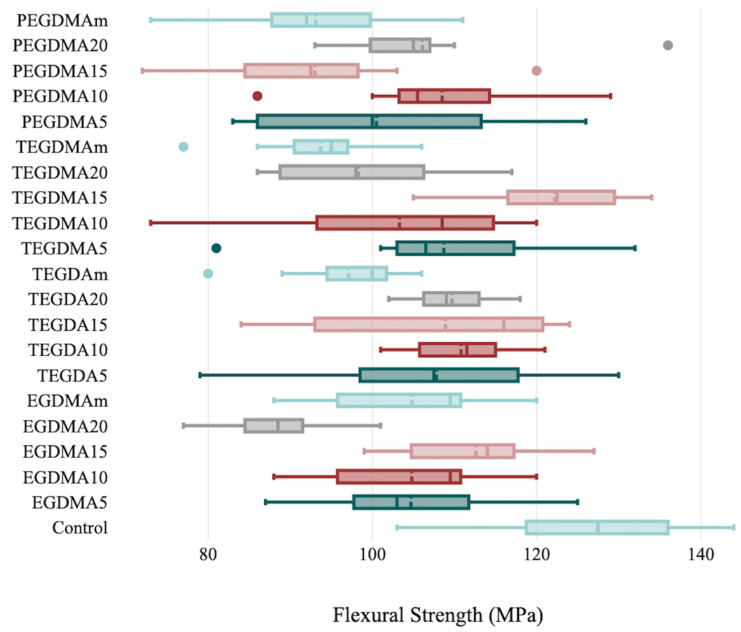
Flexural strength evaluation parameters and standard deviation bars for the control and cross-linking groups.

**Figure 4 polymers-15-02387-f004:**
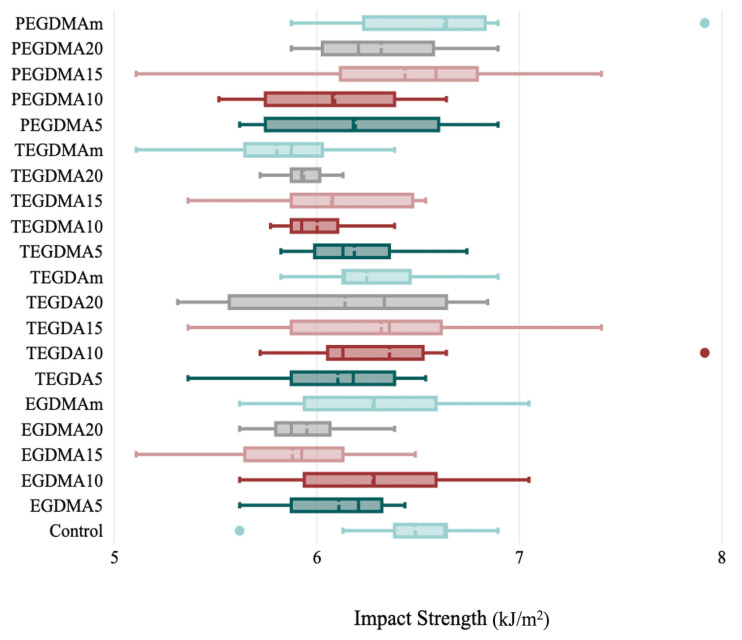
Impact strength evaluation parameters and standard deviation bars for the control and cross-linking groups.

**Figure 5 polymers-15-02387-f005:**
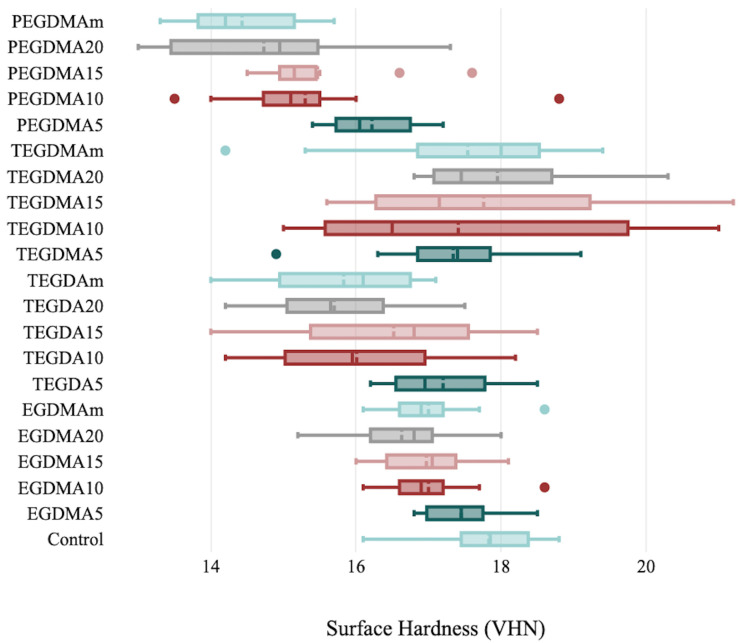
Surface hardness evaluation parameters and standard deviation bars for the control and cross-linking groups.

**Table 1 polymers-15-02387-t001:** Materials used in this study.

Name	Manufacturer	Abbreviation	Batch Number	Chemical Property	Molecular Weight (g/mol)
**Meliodent**	Heraeus Kulzer, Hanau, Germany	**M**	012294	Heat Cure Acrylic Resin	198.22 g/mol^−1^
**Methylmethacrylate**	Sigma-Aldrich, Steinheim	**MMA**	55909	Monomer	100.12 g/mol^−1^
**Ethylene glycol dimethacrylate**	Merck, Darmstadt, Germany	**EGDMA**	818847/0250	Cross-linking Agent	198.22 g/mol^−1^
**Tetraethylene glycol diacrylate**	Sigma-Aldrich, Steinheim	**TEGDA**	398802	Cross-linking Agent	302.32 g/mol^−1^
**Tetraethylene glycol dimethacrylate**	Sigma-Aldrich, (Fluka) Steinheim	**TEGDMA**	86680	Cross-linking Agent	330.37 g/mol^−1^
**Polyethylene glycol dimethacrylate**	Sigma-Aldrich, Steinheim	**PEGDMA**	409510	Cross-linking Agent	550 g/mol^−1^

**Table 2 polymers-15-02387-t002:** Definitions of measurement variables.

	n	Min	Max	Mean ± SD	*p*
Elastic Modulus	210	1939.50	3726.82	3071.88 ± 356.66	0.034
Flexural Strength	210	71.60	135.86	104.51 ± 13.76	0.734
Impact Strength	210	5.11	7.92	6.18 ± 0.46	0.024
Surface Hardness	210	13.00	21.20	16.59 ± 1.54	0.718

Kolmogorov–Smirnov Test, SD: Standard Deviation.

**Table 3 polymers-15-02387-t003:** Descriptive statistics by groups.

		Elastic Modulus *			Flexural Strength **			Impact Strength *			Surface Hardness **		
**Groups**	%	Mean ± SD	χ2	*p* ^a^	Mean ± SD	F	*p* ^a^	Mean ± SD	χ2	*p* ^a^	Mean ± SD	F	*p* ^a^
**M**		3374.13 ± 108.370			117.89 ± 6.680			6.49 ± 0.384			17.83 ± 0.800		
	5	3196.41 ± 209.310	18.392	0.001	104.62 ± 11.654	7.187	0.0001	6.11 ± 0.279	7.770	0.100	17.45 ± 0.536	1.728	0.161
	10	3189.32 ± 240.757			104.72 ± 11.576			6.28 ± 0.438			17.00 ± 0.741		
**EGDMA**	15	3378.06 ± 195.915			112.48 ± 8.941			5.88 ± 0.383			16.97 ± 0.657		
	20	2721.90 ± 368.983			88.51 ± 7.252			5.95 ± 0.231			16.63 ± 0.801		
	m10	3189.32 ± 240.757			104.72 ± 11.576			6.28 ± 0.438			17.00 ± 0.741		
	5	3182.07 ± 388.092	16.487	0.002	107.71 ± 15.885	2.371	0.067	6.10 ± 0.357	0.880	0.927	17.20 ± 0.845	2.838	0.035
	10	3193.30 ± 155.631			110.80 ± 6.147			6.36 ± 0.621			16.01 ± 1.260		
**TEGDMA**	15	3235.03 ± 439.363			109.07 ± 16.483			6.32 ± 0.594			16.52 ± 1.463		
	20	3157.30 ± 167.349			109.62 ± 5.575			6.14 ± 0.606			15.70 ± 0.979		
	m10	2691.71 ± 251.567			97.09 ± 7.857			6.25 ± 0.321			15.83 ± 1.122		
	5	3288.76 ± 228.717	19.024	0.001	108.84 ± 13.554	8.823	0.0001	6.18 ± 0.289	7.186	0.126	17.34 ± 1.195	0.207	0.933
	10	3033.73 ± 285.522			103.33 ± 15.640			6.00 ± 0.193			17.41 ± 2.415		
**TEGDA**	15	3334.06 ± 188.939			122.26 ± 9.051			6.07 ± 0.426			17.76 ± 1.985		
	20	2963.93 ± 330.215			98.18 ± 10.767			5.93 ± 0.112			17.95 ± 1.156		
	m10	2908.15 ± 329.223			93.67 ± 8.250			5.80 ± 0.373			17.54 ± 1.706		
	5	3179.54 ± 167.796	23.025	0.0001	100.58 ± 15.162	3.043	0.027	6.19 ± 0.467	6.462	0.167	16.22 ± 0.637	4.090	0.007
	10	3190.05 ± 310.729			108.53 ± 12.864			6.09 ± 0.414			15.30 ± 1.434		
**PEGDMA**	15	2699.19 ± 412.407			92.86 ± 13.064			6.44 ± 0.663			15.47 ± 0.930		
	20	2811.04 ± 269.901			106.10 ± 11.731			6.32 ± 0.375			14.73 ± 1.384		
	m10	2592.52 ± 299.765			93.06 ± 12.481			6.63 ± 0.566			14.43 ± 0.814		

* Kruskal Wallis Test, ** ANOVA, SD: Standard Deviation.

## Data Availability

The data presented in this study are available in Appendix A.

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
