# Peer review of "The Effects of Cross-Linking Agents on the Mechanical Properties of Poly (Methyl Methacrylate) Resin"

_polymers, 2023, doi:10.3390/polym15102387_

Round 1

Reviewer 1 Report

The authors disclosed the effects of cross-linkers on the mechanical properties of PMMA in this manuscript. The topic is important and the study is well-designed. This work might be acceptable after addressing the following concern: The polymerization of these PMMA synthesis seems to have low conversion. It is hard to draw reliable conclusion with the properties of these cross-linked polymers with low conversion of monomers. How were the purification of these polymers performed?

The English is good in this manuscript.

Author Response

Dear Reviewer,

We sincerely appreciate the reviewer's comments and suggestions, which have greatly helped improve the manuscript's quality. We have corrected and modified the manuscript according to the comments given by the reviewer. The authors have approved the revisions. The changes are visible in the original paper "track changes on" mode. The responses are given below.

Point 1: The authors disclosed the effects of cross-linkers on the mechanical properties of PMMA in this manuscript. The topic is important and the study is well-designed. This work might be acceptable after addressing the following concern: The polymerization of these PMMA synthesis seems to have low conversion. It is hard to draw reliable conclusion with the properties of these cross-linked polymers with low conversion of monomers. How were the purification of these polymers performed?

Response 1: We thank the reviewer for this valuable contributions. We would like to state that we are trying to complete the missing information about the purification of PMMA with the information we have added to the Materials and Methods section about the polymerization and experiment process. We contributed to the purification process by applying the ISO-specified procedures for the denture base polymers. These are the following: Using the long-term boiling method in water for polymerization, keeping the muffles in water for 24 hours until the water cools down, keeping the samples in the incubator for 48 hours before the experiment, and finally drying them at room condition for 1 hour.

We hope to have addressed the reviewer’s concerns satisfactorily.

Sincerely

Reviewer 2 Report

Comments

1. The introduction should indicate not only the purpose, but also the objectives of the study.

2. In the experimental part, the conditions for obtaining polymer samples are not given. An appropriate piece of text should be included in this section.

3. Throughout the text, it is better to give, in addition to the chemical names, the chemical formulas of the compounds. The article refers to the chemistry of polymers! In addition, the article can be improved by writing polymerization reaction schemes in the presence of cross-linking agents.

4. The literary references to the earlier conducted researches in the specified area given to the article reflect the relevance of the work. Literary references should preferably be given for the last 5 years. In this article, most of the references refer to works performed much earlier (12 out of 19). This needs to be fixed.

5. Notes on the design of the work:

Ø  Figure captions (1-3) are in very small print, you need a magnifying glass to see them.

Ø  The changing parameters in figures 1-3 must be located on the y-axis.

Ø  It is not clear why it is necessary to enter abbreviations of words repeatedly in the text of the article? Usually once at the beginning of the text is enough. In an article, for example, the abbreviation ethylene glycol dimethacrylate (EGDMA) is introduced on the first, third, and eighth pages. This also applies to other compounds. This needs to be fixed.

Author Response

Dear Reviewer,

We sincerely appreciate the reviewer's comments and suggestions, which have greatly helped improve the manuscript's quality. We have corrected and modified the manuscript according to the comments given by the reviewer. The authors have approved the revisions. The changes are visible in the original paper "track changes on" mode. The responses are given below.

Point 1: The introduction should indicate not only the purpose, but also the objectives of the study.

Response 1: We thank the reviewer for this valuable contribution. As your concern, we added the study's objectives to the last part of the Introduction section.

Point 2: In the experimental part, the conditions for obtaining polymer samples are not given. An appropriate piece of text should be included in this section.

Response 2: We would like to thank the reviewer for this important criticism. We have detailed the missing information about the preparation of the test samples and the polymerization process. We amended the Materials and Methods section based on the recommendation.

Point 3: Throughout the text, it is better to give, in addition to the chemical names, the chemical formulas of the compounds. The article refers to the chemistry of polymers! In addition, the article can be improved by writing polymerization reaction schemes in the presence of cross-linking agents.

Response 3: We made the necessary adjustments for the important deficiency stated by the reviewer. The chemical formulas of the components used in the study, the schematic representation of the synthesized structures and how the crosslinker is attached to the main polymer structure are given in Figure 1.

Point 4: The literary references to the earlier conducted researches in the specified area given to the article reflect the relevance of the work. Literary references should preferably be given for the last 5 years. In this article, most of the references refer to works performed much earlier (12 out of 19). This needs to be fixed.

Response 4: We agree with the critic in this comment. For this reason, we specifically stated that among the limitations of our study, there needs to be a sufficient number of studies similar to the one we have done recently. As the reviewer noted, we have updated the references section by referring to more recent literature.

Point 5: Notes on the design of the work:

Ø  Figure captions (1-3) are in very small print, you need a magnifying glass to see them.

Ø  The changing parameters in figures 1-3 must be located on the y-axis.

Ø  It is not clear why it is necessary to enter abbreviations of words repeatedly in the text of the article? Usually once at the beginning of the text is enough. In an article, for example, the abbreviation ethylene glycol dimethacrylate (EGDMA) is introduced on the first, third, and eighth pages. This also applies to other compounds. This needs to be fixed. 

Response 5:

Ø  We have recreated the figure texts with a design that will make them more legible with the reviewer's interpretation.

Ø  We have rearranged the x-y axis describing the data in the figures as specified by the reviewer.

Ø  We are sorry for the text error you specified. The sentences containing the components abbreviations given in the text have been revised, and the mistakes have been corrected.

We hope to have addressed the reviewer’s concerns satisfactorily.

Sincerely

Reviewer 3 Report

In this manuscript Ceylan and coworkers studied the effect of cross-linkers addition on the mechanical properties of PMMA resin prepared for dentures. The study is well conducted, a large library of samples has been schematically prepared and tested following the international standars for denture polymers.

The addition of bifunctional acrylates in the polymerization of methyl methacrylate has been attempted several times and many works are present in the literature, however, concerning PMMA for denture there are only few examples with few variety in samples and crosslinking agents.

This study is very interesting due to the large amount of sample prepared specifically for dentures using 4 differents cross-linkers at different concentrations, giving the reader and overview on the achiviable effect on the mechanical properties.

I recomend this work for publication in Polymers after a minor revision addressing the subsequent indications:

-Conclusions must be rewritten in a discorsive way.

-In point 3 of the conclusions and in the abstract is stated that the addition of cross-linkers improved the mechanical properties of the materials. I disagree, it is true that mechanical properties changes, but I don't see any real improvment, almost always there is a worsening, and moreover is often observed a drop in reproducibility. My suggestion is to state that by observing these result and selecting the right cross-linker it is possible to plan a tuning of the mechanical properties in the desired direction.

Author Response

Dear Reviewer,

We sincerely appreciate the reviewer's comments and suggestions, which have greatly helped improve the manuscript's quality. We have corrected and modified the manuscript according to the comments given by the reviewer. The authors have approved the revisions. The changes are visible in the original paper "track changes on" mode. The responses are given below.

Point 1: In this manuscript Ceylan and coworkers studied the effect of cross-linkers addition on the mechanical properties of PMMA resin prepared for dentures. The study is well conducted, a large library of samples has been schematically prepared and tested following the international standars for denture polymers.

The addition of bifunctional acrylates in the polymerization of methyl methacrylate has been attempted several times and many works are present in the literature, however, concerning PMMA for denture there are only few examples with few variety in samples and crosslinking agents.

This study is very interesting due to the large amount of sample prepared specifically for dentures using 4 differents cross-linkers at different concentrations, giving the reader and overview on the achiviable effect on the mechanical properties.

I recomend this work for publication in Polymers after a minor revision addressing the subsequent indications:

-Conclusions must be rewritten in a discorsive way.

-In point 3 of the conclusions and in the abstract is stated that the addition of cross-linkers improved the mechanical properties of the materials. I disagree, it is true that mechanical properties changes, but I don't see any real improvment, almost always there is a worsening, and moreover is often observed a drop in reproducibility. My suggestion is to state that by observing these result and selecting the right cross-linker it is possible to plan a tuning of the mechanical properties in the desired direction.

Response 1: Thank you for your feedback. We agree with your assessment that the addition of cross-linkers to conventional PMMA denture base material can result in both improvements and worsening of mechanical properties. The type and concentration of cross-linker used is important, as is the method of polymerization. In our study, we found that the addition of cross-linkers at concentrations between 5% and 15% improved the flexural strength, elastic modulus, and impact strength of the material. However, we also found that adding PEGDMA at concentrations ranging from 5% to 20% significantly decreased the surface hardness of the material. These findings suggest that the addition of cross-linkers to conventional PMMA denture base material can be used to tune the mechanical properties of the material in the desired direction. However, it is important to carefully select the type and concentration of cross-linker used, as well as the method of polymerization.

The conclusion section has been rewritten according to the reviewer's suggestions.

We hope to have addressed the reviewer’s concerns satisfactorily.

Sincerely
